# Thermodynamic Profiling Reveals DNA Polymerase Template Binding, Substrate Incorporation, and Exonuclease Function

**DOI:** 10.3390/ijms262411909

**Published:** 2025-12-10

**Authors:** Yaping Sun, Wu Lin, Kang Fu, Jie Gao, Xianhui Zhao, Yun He, Hui Tian

**Affiliations:** Research Center of Molecular Diagnostics and Sequencing, Research Institute of Tsinghua University in Shenzhen, Shenzhen 518000, China; sunyp@tsinghua-sz.org (Y.S.);

**Keywords:** isothermal titration calorimetry, Phi29 DNA polymerases, thermodynamic profiling, exonuclease activity, template binding, incorporation

## Abstract

Isothermal titration calorimetry (ITC) provides direct insight into the energetics of DNA polymerase function, including binding, catalysis, and exonuclease activity. We characterized a Phi29 mutant polymerase (SS_01) engineered to incorporate non-natural nucleotides in the presence of Mg^2+^, a function absent in the wild-type enzyme. ITC analyses revealed that SS_01 binding to the primed template was strongly influenced by metal ions. In the presence of Mg^2+^, the polymerase displayed tight binding (KD = 243 nM) and a clear exothermic signal, indicating activation of a large fraction of catalytically competent molecules. By contrast, in the presence of Ca^2+^, binding produced weaker exothermic signals (KD = 317 nM), suggesting less efficient binding complex formation. During dNTP- or oligonucleotide-tagged dNTP-driven polymerization, ITC profiles with Mg^2+^ exhibited pronounced endothermic heat changes, whereas with Ca^2+^, only minimal heat changes were observed. When binding only oligonucleotide-tagged dNTPs, the polymerases showed distinct thermodynamic behavior: in the presence of Mg^2+^, high substrate concentrations induced endothermic responses, while in the absence of catalytic ions, binding remained exothermic. Exonuclease activity monitored using unmodified oligonucleotides yielded strong exothermic signals in the presence of Mg^2+^ but weak responses in the presence of Ca^2+^, confirming strict ion dependence. Together, these data demonstrate that ITC directly captures the metal ion-dependent energetics of SS_01, providing mechanistic insight into its polymerization and exonuclease functions.

## 1. Introduction

Isothermal amplification DNA polymerases play a central role in DNA replication and repair. In recent years, engineered isothermal amplification DNA polymerases for robust strand displacement, high processivity, and high fidelity using unnatural or chemically modified nucleotides have also been widely utilized in synthetic biology and biotechnology applications, such as expanded genetic systems, XNA synthesis, and nanopore-coupled sequencing [1,2,3,4,5]. Despite advances in kinetic and structural studies, the thermodynamic basis underlying natural or unnatural nucleotide interactions with DNA polymerase or polymerase-primed templates remains poorly understood.

The catalytic activity of DNA polymerases critically depends on the presence of divalent metal ions, most commonly magnesium (Mg^2+^) [6], which facilitate phosphoryl transfer by stabilizing the transition state and coordinating the triphosphate moiety of the incoming nucleotide [7]. However, not all divalent metal ions possess catalytic activity. For example, calcium (Ca^2+^) can bind within the active site of DNA polymerases, but fail to support phosphoryl transfer due to suboptimal ionic radius and geometry, thereby trapping the polymerase in an inactive conformation [8,9]. Such catalytic and non-catalytic ion conditions not only determine enzymatic turnover but also influence the conformational dynamics and binding affinity between polymerases and DNA. Understanding these ion-dependent effects is crucial for both mechanistic enzymology and the rational engineering of polymerases for specialized applications.

Oligonucleotide-tagged nucleotides, as a kind of unnatural substrates, have been widely used in polymerase-nanopore sequencing system to modulate strand translocation and sequencing signal generation [1]. Single-stranded DNA has been shown to engage the 3′-5′ exonuclease domain of DNA polymerases for degradation [10,11]. A detailed thermodynamic understanding for these interactions from strand displacing DNA polymerases remains lacking.

Isothermal titration calorimetry (ITC) is a powerful biophysical technique that enables the label-free and direct measurement of molecular interactions under near-physiological conditions. It provides key thermodynamic parameters, including dissociation constants (KD) and enthalpy changes (ΔH), offering mechanistic insight into the driving forces of complex formation [12,13]. While ITC has been extensively applied in the study of protein–ligand and protein–nucleic acid interactions [14], its application to DNA polymerase systems has been relatively limited, particularly in the context of substrate binding and incorporation involving natural or unnatural nucleotides. ITC assays for Taq/Klentaq DNA polymerases showed the optimal temperatures (40–50 °C) for binding with DNA [15].

In this study, we investigated the binding and catalytic properties of SS_01, a mutant of the bacteriophage Phi29 DNA polymerase, which we engineered to incorporate unnatural nucleotides in the presence of Mg^2+^—a capability absent in the wild-type enzyme. Using ITC, rolling circle amplification (RCA), and gel-based binding assays, we systematically examined the thermodynamic behavior of SS_01 with primed template complexes, single-stranded DNA, and oligonucleotide-tagged dNTP analogs. Our results reveal distinct binding modes, shedding light on the mechanistic basis for SS_01’s exonuclease activity and incorporation activity, and providing new insights for the design of polymerases compatible with modified nucleotides in nanopore sequencing and other biotechnological applications.

## 2. Results

### 2.1. Binding Affinity

To explore the thermodynamic underpinnings of DNA polymerase activity, we engineered a mutant of the Phi29 DNA polymerase (SS_01, M97K, A484E, K512Y) that is capable of incorporating oligonucleotide-tagged dNTPs in the presence of Mg^2+^—a capability that is absent in the wild-type Phi29 enzyme (Figure 1A,B). Oligonucleotide-tagged dNTPs are commonly used as unnatural substrates instead of dNTPs in long-read nanopore sequencing systems [1]. We proposed that the three substitutions collectively reshaped both the primer-binding interface and the catalytic microenvironment (Figure 1A). The M97K substitution likely enhanced electrostatic interactions with the primer terminus, stabilizing the primer–template junction during nucleotide incorporation. The A484E mutation, located proximal to the catalytic center, may strengthen coordination with the Mg^2+^ ion and promote a more favorable geometry for metal-assisted catalysis. Meanwhile, K512Y, positioned near the substrate entry channel, may modulate the accessibility and accommodation of structurally diverse nucleotide analogs. Isothermal titration calorimetry (ITC) assays were employed to measure the thermodynamic changes.

#### 2.1.1. Binding Affinity Under Catalytic and Non-Catalytic Metal Ion Conditions

Firstly, we examined the binding affinity between the polymerase (SS_01) and a primed template complex in the presence and absence of Mg^2+^ using ITC experiments (Figure 1C). The results of three independent replicates consistently showed that the dissociation constant (KD) in the presence of Mg^2+^ was approximately 243 nM, whereas in the absence of Mg^2+^, the KD value was about 129 nM (Figure 1D; Appendix A, showing the raw ITC data for all three replicate measurements; Appendix A). At first glance, these results suggest that the enzyme binds the primed template more tightly in the absence of Mg^2+^. However, this interpretation is complicated by the fact that metal ions such as Mg^2+^ are required for the DNA polymerase to adopt its catalytically competent, active conformation.

Previous studies have demonstrated that DNA polymerases require divalent metal ions to form a stable ternary complex with both the primed template and the incoming nucleotide [16,17]. Without Mg^2+^, the enzyme largely remains in an inactive or partially folded state, with only a small fraction capable of binding the template efficiently. Thus, under Mg^2+^-free conditions, even a low concentration of the primed template is sufficient to saturate the limited number of binding-competent enzyme molecules, resulting in an apparently lower KD value. In contrast, the presence of Mg^2+^ promotes the active conformation in a larger proportion of the polymerase population, increasing the total number of primed template-binding sites and thereby requiring a higher concentration of DNA to reach saturation. Consistent with the interpretation, gel-based binding assays showed that DNA polymerases bound strongly to primed templates in the presence of Mg^2+^, whereas weak binding was observed under non-catalytic conditions (Figure 1E,F). Therefore, when comparing the interactions between DNA polymerases and primed template under catalytic and non-catalytic metal ion conditions, a higher KD value in the presence of Mg^2+^ may not indicate weaker binding but rather reflect a greater proportion of enzyme molecules actively engaging the primed template, leading to an overall increase in binding capacity.

#### 2.1.2. Binding Affinity Under Various Metal Ion Conditions

To further test the binding affinity of SS_01 in different catalytic ion conditions, we also incubated polymerases and primed templates with Mg^2+^, Mn^2+^, Ca^2+^, and Sr^2+^. These metal ions possess distinct ionic radii: Mg^2+^ (0.72 Å), Mn^2+^ (0.83 Å), Ca^2+^ (1.00 Å), and Sr^2+^ (1.18 Å) [18]. Mg^2+^ and Mn^2+^ serve as primary catalytic cofactors for DNA polymerases due to their optimal size for precise transition-state stabilization in phosphoryl transfer reactions. Ca^2+^ and Sr^2+^—with significantly larger radii—act as negative controls, as their geometric mismatch disrupts active-site coordination and prevents catalysis. Ca^2+^ and Sr^2+^ were often used in nanopore sequencing systems to produce stagnation signals [19,20]. Our gel results showed that Mg^2+^ and Mn^2+^ facilitated the binding affinity of the primed template and polymerases, while Ca^2+^ and Sr^2+^ only showed weak function for the binding affinity (Figure 1E,F). Then, to test the thermodynamics changes in the various divalent metal ions, ITC assays were performed in the presence of Mg^2+^ or Ca^2+^. Its results also showed that the KD value in the presence of Mg^2+^ was smaller than that in the presence of Ca^2+^ (317 nM) in agreement with the gel-based binding assays (Figure 1G; Appendix A, showing the raw ITC data for all three replicate measurements; Appendix A). This indicates that when comparing different catalytic metal ion conditions, a lower KD value refers to a higher binding affinity between the polymerase and the primed template.

To verify that the ITC-derived titration curves and the measured values indeed reflect specific binding, we performed two negative control experiments: titrating the reaction buffer into the polymerases and titrating the primed templates into the reaction buffer (Appendix A showing the raw ITC data for all two replicate measurements). Each control was repeated twice, yielding consistent results. In all cases—whether in the absence of catalytic ions or in the presence of Mg^2+^ or Ca^2+^—no discernible titration curves were observed, in stark contrast to the experimental group.

### 2.2. The Thermodynamic Changes During the Incorporation of SS_01 Using dNTPs

To evaluate the polymerization activity of SS_01 under different metal ion conditions, we performed rolling circle amplification (RCA) assays using single strand circular DNA (SSC) templates and dNTPs. The results showed that SS_01 efficiently catalyzed DNA synthesis in the presence of Mg^2+^ and Mn^2+^, but not in the presence of Ca^2+^ and Sr^2+^ (Figure 2A,B). Based on this observation, Mg^2+^ and Ca^2+^ were selected as representative conditions for active and inactive polymerization, respectively, to further investigate the thermodynamic profiles of the incorporation process.

The ITC results revealed that after primed templates binding with SS_01 polymerases in the presence of Ca^2+^, the titration of dNTPs resulted in only minimal heat changes, indicating an absence of productive DNA synthesis (Figure 2C,D; Appendix A, showing the raw ITC data for all three replicate measurements). In contrast, once the enzyme-primed template complex was formed in the presence of Mg^2+^, the introduction of dNTPs triggered active DNA chain elongation, resulting in the endothermic heat profile (Figure 2C,D; Appendix A, showing the raw ITC data for all three replicate measurements). We suggested that the high endothermic heat profile may be from three parts: (1) the catalytic incorporation of nucleotides; (2) the substantial conformational changes in polymerases, (3) the translocation of the DNA templates.

To further validate the reliability of our ITC data, we performed an additional titration control in which the reaction buffer was titrated into the pre-formed enzyme-primed template complex. In the presence of Mg^2+^, the reaction buffer titration produced no titration points, a result entirely distinct from the clear incorporation isotherms observed when titrating dNTPs (Appendix A, showing the raw ITC data for all two replicate measurements). This control confirms that the ITC signals obtained in our experimental groups arise from genuine incorporation events rather than non-specific dilution.

### 2.3. The Thermodynamic Changes During the Incorporation of SS_01 Using Oligonucleotide-Tagged dNTP Analogs

To further explore the thermodynamic behavior of polymerase activity, we examined the interaction between SS_01 and oligonucleotide-tagged dNTP analogs (Figure 3A), which were synthesized according to the previous references [1]. These modified substrates, consisting of short oligonucleotide moieties attached to the nucleotide base, are structurally bulky.

#### 2.3.1. Thermodynamic Characterization of SS_01 Binding to Oligonucleotide-Tagged dNTPs Under Catalytic and Non-Catalytic Metal Ion Conditions

Prior to assessing polymerization activity, we evaluated the binding affinity between SS_01 and the oligonucleotide-tagged dNTPs in the presence or absence of divalent metal ions (Figure 3A). Since we had already included the reaction buffer-into-SS_01 titration as a negative control in the polymerase-primed template binding experiments (Appendix A), for the assays involving the oligonucleotide-tagged dNTPs, we performed the complementary control experiments of titrating the oligonucleotide-tagged dNTPs into the reaction buffer. Under all tested conditions—without catalytic ions, or in the presence of Ca^2+^ or Mg^2+^—they produced no detectable titration points (Appendix A showing the raw ITC data for both replicate measurements). Each control was repeated twice, yielding consistent results. ITC analysis revealed that the binding of SS_01 to oligonucleotide-modified dNTPs was consistently exothermic across a range of substrate concentrations (10 μM, 20 μM, and 100 μM) in the absence of catalytic ion Mg^2+^ (Figure 3B; Appendix A, showing the raw ITC data for all three replicate measurements). Interestingly, in the presence of Mg^2+^ ions, which promote catalysis, the thermodynamic profiles shifted significantly. At 10 μM, the interaction produced minimal heat changes, with no clear exothermic or endothermic trends. However, at higher concentrations (20 μM and 100 μM), the reaction profile became distinctly endothermic, indicating altered binding dynamics or possible initiation of low-level exonuclease activity or structural rearrangements of the enzyme-substrate complex (Figure 3C; Appendix A, showing the raw ITC data for all three replicate measurements). These results suggest that the presence of catalytic metal ions and the concentration of modified substrates together influence the energetics and potentially the mode of interaction between SS_01 and oligonucleotide-tagged dNTPs.

This binding behavior differed markedly from that observed with double-stranded DNA templates. In the presence of identical Mg^2+^, the binding of SS_01 to a 10 μM primed template produced a clear exothermic signal (Figure 1D and Appendix A), whereas binding to 10 μM oligonucleotide-tagged dNTPs resulted in minimal heat changes, with no clear exothermic or endothermic trends (Figure 3C and Appendix A). This contrast suggests that SS_01 interacts differently with single-stranded unnatural substrates compared to primed template. This behavior may be attributed to the structural and functional characteristics of the modified substrates. We propose that two factors are therefore likely to account for the observed thermodynamic differences between enzyme binding to oligonucleotide-tagged dNTPs or primed templates: (i) the difference in DNA structural type, and (ii) the excessively high DNA concentration (20 μM or 100 μM).

To test this hypothesis, ITC assays were performed using high concentrations of primed template to detect the binding affinity with SS_01. When 100 μM primed template was used in the presence of Mg^2+^ conditions, the binding reaction revealed exothermic changes (Appendix A). This behavior was markedly different from the binding profile observed for 100 μM oligonucleotide-tagged dNTPs under the same conditions (Figure 3C and Appendix A). Therefore, high concentration is not the reason for the difference in thermodynamic changes for binding SS_01 with primed template or oligonucleotide-tagged dNTPs.

To further understand the differential binding behavior of SS_01 with various structural types of DNA, we analyzed available structural data from the Protein Data Bank (PDB). Specifically, we obtained two crystal structures of the Phi29 DNA polymerase in complex with primed template and incoming dNTPs (2PYL and 2PZS) [11], as well as two structures of Phi29 DNA polymerase bound to single-stranded DNA (1XHZ and 1XI1) [10,21] (Figure 4). Remarkably, all two ssDNA-bound structures consistently showed the single-stranded DNA occupying the 3′-5′ exonuclease domain, regardless of the presence or absence of Mg^2+^ ions.

#### 2.3.2. Thermodynamic Characterization of SS_01 Degrading the Oligonucleotides in the Presence of Mg^2+^

To explore the 3′-5′ exonuclease activities of SS_01, we incubated two kinds of oligonucleotides with SS_01 polymerases: unmodified and 3′-protected phosphorothioate (PS) oligonucleotides (oligos). Phosphorothioate modifications have been known to effectively inhibit exonuclease activity, as reported for *Vibrio parahaemolyticus* phage VpV262 DNA polymerases, *Psychrobacillus* PB DNA polymerases and some psychrophilic polymerases [22,23,24,25,26]. We quantified the fraction of oligonucleotides remaining after the reaction. A lower proportion of remaining oligonucleotides corresponds to higher exonuclease activity. As shown in Figure 5A–C, in the presence of Mg^2+^ or Mn^2+^, the DNA polymerase efficiently degraded the unmodified oligonucleotides but failed to degrade the PS-modified oligonucleotides. In contrast, in the presence of Ca^2+^, the DNA polymerase was unable to degrade either the modified or the unmodified oligonucleotides.

To clarify the thermodynamic change in the degradation process for exonuclease activities of SS_01, we also performed the ITC assays using SS_01 and unmodified oligonucleotides in the presence Mg^2+^ or Ca^2+^ (Figure 5D). Comparison with the condition lacking catalytic ions, a pronounced exothermic reaction was observed in the presence of Mg^2+^ or Ca^2+^, indicating interactions between the oligonucleotides and SS_01 polymerase (Figure 5E,F; Appendix A showing the raw ITC data for all three replicate measurements.). This suggests that the oligonucleotides entered the exonuclease domain of the enzyme. Intriguingly, the heat change was markedly greater in the presence of Mg^2+^ than in the presence of Ca^2+^ (Figure 5E,F and Appendix A). An additional control experiment was performed in which the unmodified oligonucleotides were titrated into the reaction buffer in the presence of Mg^2+^. The conditions produced no detectable titration points (Appendix A), which is in clear contrast to the experimental conditions and further supports that the oligonucleotides degradation was an exothermic process.

All of these results suggest that the catalytic metal ions are required not only for polymerization but also for activating exonuclease function.

#### 2.3.3. Thermodynamic Characterization of the Incorporation of SS_01 Using Oligonucleotide-Tagged dNTPs

We investigated oligonucleotide-tagged dNTPs incorporation in the presence or absence of Mg^2+^ through ITC measurements (the workflow similar to that in Figure 2C). The results revealed that, in the absence of catalytic ions, addition of oligonucleotide-tagged dNTPs after the polymerase-primed template complex formed, caused only minimal heat changes (Figure 6A,B; Appendix A, showing the raw ITC data for all three replicate measurements). By contrast, in the presence of Mg^2+^ (Figure 6A,C), oligonucleotide-tagged dNTPs addition produced an endothermic signal, which was consistent with the pattern observed for dNTP-driven synthesis (Figure 2D).

Since in the above binding experiments, we observed that using 5 μM SS_01 polymerase with 100 μM oligonucleotide-modified substrates resulted in a clear endothermic response in the presence of Mg^2+^ (Figure 3C and Appendix A), we sought to determine whether the observed endothermic heat during DNA synthesis with double-stranded DNA templates and oligonucleotide-modified dNTP substrates arised from the direct interaction between the polymerase and the modified dNTPs. To address this, we performed additional control experiments. Namely, only three kinds of substrates were provided, with the missing one corresponding to the first nucleotide required for template-directed synthesis, which resulted in failing the initiation of DNA replication. In the three-substrate condition, an endothermic response was still observed, but the heat change was markedly attenuated compared with the complete synthesis condition (Figure 6A,D and Appendix A). We propose that, when the full set of substrates is absent, the polymerase is unable to initiate DNA synthesis. Under such circumstances, the high concentration of oligonucleotide-modified dNTP likely enters the 3′-5′ exonuclease active center of the polymerase. A similar effect was seen under the condition containing only polymerase, oligonucleotide-modified dNTP substrates, and Mg^2+^ (Figure 3C), where the oligonucleotide-modified substrates predominantly entered the exonuclease domain of the enzyme. Therefore, our findings support the notion that the incorporation of SS_01 using oligonucleotide-tagged dNTPs was an endothermic reaction.

## 3. Discussion

In this study, we characterized a Phi29 DNA polymerase mutant (SS_01) that showed a distinct biochemical behavior compared to the wild-type enzyme. While the wild-type Phi29 DNA polymerases strictly required canonical nucleotides for DNA synthesis, SS_01 was able to incorporate unnatural substrates in the presence of Mg^2+^. In addition, SS_01 also retained strong 3′-5′ exonuclease proofreading activity and potent strand displacement ability, which underpinned its utility in isothermal amplification techniques [27]. We employed ITC assays to investigate the thermodynamic profiles of SS_01 DNA polymerases. Our findings reveal distinct heat signatures associated with DNA binding, DNA synthesis and exonuclease activity, providing deeper insights into the enzyme’s functional dynamics.

### 3.1. Metal Ion Dependence of Polymerase-Primed Templates Binding

Our ITC measurements showed that SS_01 binding to the primed template was an exothermic process at 25 °C, similar to the exothermic DNA-binding behavior reported for the Y-family lesion-bypass polymerase Dpo4 from *Sulfolobus solfataricus* at 60 °C [28]. The dissociation constants (KD) and gel-based assays for SS_01 binding to primed templates indicated that, when comparing DNA polymerase-primed template interactions under catalytic versus non-catalytic metal ion conditions, a higher KD value in the presence of catalytic ions did not necessarily imply weaker binding. Instead, it likely reflected a larger fraction of enzyme molecules adopting an active conformation and engaging with the primed template, thereby increasing the overall binding capacity. This interpretation is consistent with the requirement of Mg^2+^ for DNA polymerases to assume a catalytically competent state [29,30]. However, under different catalytic metal ion conditions, a lower KD value corresponded to stronger polymerase-primed template interactions. Specifically, Mg^2+^ markedly enhanced SS_01 binding affinity, yielding lower KD values, whereas Ca^2+^ showed only weak binding with a higher KD value. These findings align with the established roles of Mg^2+^ as a primary catalytic cofactor in DNA synthesis, while the larger ionic radius of Ca^2+^ interferes with optimal active-site coordination, thereby reducing its effectiveness [31,32].

### 3.2. Polymerization Activity and Thermodynamic Profiles

Rolling circle amplification (RCA) assays demonstrated that SS_01 efficiently catalyzed DNA synthesis using dNTPs in the presence of Mg^2+^ and Mn^2+^, but not in the presence of Ca^2+^ and Sr^2+^. Protein structure studies of DNA polymerase Pol η have shown that, compared with Mg^2+^-bound complexes, Ca^2+^ binding altered the geometry of the catalytic center, causing the catalytic residue Arg61 to adopt misaligned orientations relative to the incoming nucleotide [17]. This structural deviation helps explain the impaired catalytic efficiency observed under Ca^2+^ conditions. In line with this, ITC revealed negligible heat change upon dNTPs addition in the presence of Ca^2+^, whereas a clear endothermic signature was observed in the presence of Mg^2+^, reflecting active polymerization, conformational adjustments, and substrate translocation. However, previous studies on the Klenow fragment of *E. coli* DNA polymerase I, which lacks 3′-5′ exonuclease activity, have reported exothermic heat changes during DNA synthesis [33]. These findings contrast with our results, highlighting the significant impact of exonuclease activity on the thermodynamic profiles of DNA polymerase reactions.

Our ITC results of SS_01 incorporation using oligonucleotide-tagged dNTPs showed minimal heat change in the absence of Mg^2+^ but pronounced endothermic signals in the presence of Mg^2+^—consistent with the incorporation using dNTPs and suggestive of synthesis-like processes or conformational modulation. Further control experiments showed that incomplete substrate mixtures (lacking one type of dNTPs) or buffer-only titrations produced attenuated or negligible heat responses, respectively, further supporting that the incorporation reaction is endothermic.

In addition, previous studies have shown that Ca^2+^ can function as a catalytic ion for archaeal *Pyrococcus abyssi* DNA Pol (PabPolB), a kind of family-B DNA polymerases, albeit with substantially reduced reaction rates [34]. Phi29 is a well-characterized member of the B-family polymerases. In our RCA and ITC assays, however, we did not detect any DNA product and pronounced heat changes indicative of DNA synthesis by SS_01 in the presence of Ca^2+^. This absence of signal does not fully exclude the possibility that Ca^2+^ may support a very slower polymerization reaction.

### 3.3. Oligonucleotide-Tagged dNTP or Single-Stranded DNA Binding with Polymerases

Our binding results demonstrate that SS_01 possessed distinct thermodynamic behaviors when binding to oligonucleotide-tagged dNTPs compared to primed templates. While the binding of primed templates consistently produced exothermic signals in the presence of Mg^2+^, bulky oligonucleotide-tagged substrates showed minimal or endothermic changes, suggesting altered enzyme conformations or partial engagement of the exonuclease domain. Structural insights from Phi29-family polymerases supported that oligonucleotides often occupied the 3′-5′ exonuclease active sites [10,21].

Exonuclease assays showed SS_01 degraded unmodified oligonucleotides in presence of Mg^2+^/Mn^2+^ but cannot process PS-modified ones; in the presence of Ca^2+^ or Sr^2+^, no degradation occurred, consistent with previous reports for related polymerases [22,23,24,25,26]. ITC assays confirmed that degradation was an exothermic process, with stronger heat release in the presence of Mg^2+^ than Ca^2+^, highlighting the essential role of catalytic ions in activating exonuclease function as well as polymerization. This is consistent with previous studies on bacteriophage T4 DNA polymerase, which demonstrated that its exonuclease activity requires two divalent metal ions [35]. One metal ion coordinated with a water molecule to generate a metal–hydroxide species that was positioned to attack the phosphodiester bond at the cleavage site. The second metal ion was proposed to stabilize the leaving 3′-hydroxyl group and to help position the O–P–O bond angles during the transition state.

Our findings were also consistent with previous studies showing that the PabPolB DNA polymerase possessed no exonuclease activity in the presence of Ca^2+^ but retains activity in the presence of Mg^2+^ [34]. We speculate that this difference may reflect a mechanism similar to that observed during nucleotide incorporation, in which Ca^2+^ and Mg^2+^ play distinct structural roles. The larger ionic radius of Ca^2+^ compared with Mg^2+^ (1.00 Å vs. 0.72 Å) [18] may result in altered spatial positioning within the active sites. Further structural studies under different ionic conditions will be necessary to fully elucidate the mechanistic basis of SS_01 exonuclease activity.

This study highlights the pivotal interplay between metal cofactors, substrate architecture, and enzyme conformation on the thermodynamics of DNA polymerase activity. The observed binding and catalytic behaviors under diverse conditions enhance mechanistic understanding and inform design strategies for engineered polymerases tailored to biotechnological applications.

## 4. Materials and Methods

### 4.1. Plasmid Construction

The coding sequences of Phi29 and SS_01 DNA polymerase were codon-optimized for *E. coli* expression, synthesized and cloned into the Pet30a plasmids by Genscript Biotechnology (Nanjing, China). All constructs included C-terminal TEV-cleavable His-tags.

### 4.2. Protein Expression

Recombinant protein expression plasmids were introduced into chemically competent *E. coli* BL21 (DE3) cells (ToloBio, Wuxi, China) following the supplier’s protocol. Briefly, competent cells were thawed on ice for 2 min, mixed with 100 ng of plasmid DNA, and incubated on ice for 30 min. Heat-shock was performed at 42 °C for 90 s using a precision metal bath (H203-100C, COYOTE, Beijing, China), followed by cooling on ice for 3 min. The cells were then supplemented with 500 μL LB medium without antibiotics and allowed to recover at 37 °C with shaking (200 rpm) for 45 min. After centrifugation, the pellet was resuspended in 200 μL LB and plated on LB agar plates (10 cm) containing kanamycin. Plates were incubated overnight at 37 °C.

Single colonies were cultured in 5 mL of LB medium supplemented with kanamycin (50 µg/mL) overnight at 37 °C. This culture was transferred into 200 mL LB medium and grown at 37 °C with agitation (200 rpm) until an OD600 value of 0.6 was achieved. Protein expression was induced with 1 mM isopropyl-D-1-thiogalactopyranoside (IPTG, A100487, Sango Biotech, Shanghai, China), and cells were incubated at 16 °C for 14 h–16 h. Harvested cells were pelleted by centrifugation at 8000 rpm for 10 min, washed twice with ice-cold PBS, and stored at −80 °C until purification.

All purification steps were performed at 4 °C. Frozen pellets were resuspended in 50 mL binding buffer (50 mM Tris-HCl (pH 7.5) and 500 mM NaCl) and lysed by sonication on ice (Scientz-IID, 01C1503, Scientz Biotechnology, Ningbo, China; 100 W, 2 s on/5 s off, total 20 min). His tagged proteins were purified using an in-house Ni Sepharose 6FF gravity column. Namely, 1.3 g of cells were processed with 3 mL of Ni Sepharose 6FF resin (17531801, Cytiva, Marlborough, MA, USA) (50% suspension) for purification. The resin was equilibrated with 10 column volumes of binding buffer (50 mM Tris-HCl (pH 7.5) and 500 mM NaCl) after rinsing with 5 column volumes of double distilled water. Protein supernatants were slowly added to the chromatography column. The column was washed with 10 column volumes of binding buffer, and the protein was eluted using an imidazole step gradient (20–300 mM) prepared in elution buffer (20 mM Tris, 500 mM NaCl). Elution fractions were examined by the homemade 10% SDS-PAGE. High-purity components then underwent 10 K ultrafiltration (157655, Millipore, Billerica, MA, USA) to remove imidazole and obtain a high-concentration protein.

For His-tag removal, 600 μg of purified protein was digested overnight at 4 °C with 400 μL TEV protease (300 ng/μL) in TEV cleavage buffer (50 mM Tris-HCl, 5 mM NaCl, 1 mM EDTA, 1 mM DTT, pH 8.0). Due to the TEV protease with His-tags, an in-house Ni Sepharose 6FF gravity column was used to remove the TEV protease and the cleaved His-tags in the reaction mixture. The proteins were further concentrated by ultrafiltration and analyzed by 10% SDS-PAGE. Protein concentrations were determined using the Pierce BCA Protein Assay Kit (23225; Thermo Fisher Scientific, Rockford, IL, USA) and further checked with the reference standard of bovine serum albumin (BSA).

Homemade SDS-PAGE were prepared using the 10% PAGE gel quick preparation kit. For protein electrophoresis, 10 μL of protein samples was mixed with 2.5 μL 6× loading buffer, boiled at 95 °C for 5 min, centrifuged at 12,000 rpm for 1 min, and loaded onto gels. Electrophoresis was performed in tris-glycine buffer at 200 V for 35 min. Gels were stained using an eStain L1 system (L00657C, Genscript, Nanjing, China).

### 4.3. Structural Analysis

Protein structures were obtained from the Protein Data Bank (PDB) [36]. The structure visualization and electrostatic potential maps of the surface were created and analyzed using PyMOL (Version 2.5.0) [37].

### 4.4. Exonuclease Activity

To evaluate exonuclease activity, we designed two oligonucleotides without modification (TCCTAACGAGATTAGTTTTGCTGTT) and with phosphorothioates (PS) modification (TCCTAACGAGATTAGTTTTGCT*G*T*T). Briefly, reactions containing 1 μM oligonucleotide and 200 nM polymerase in phi29 buffer (New England Biolabs, Ipswich, MA, USA) were incubated at 30 °C for 30 min. Products were analyzed by 10% TBE-urea PAGE (EC68752BOX, ThermoFisher, Cleveland, OH, USA) in an XCell SureLock Mini-Cell (Thermo Fisher Scientific Inc., Cleveland, OH, USA), as instructed by the manufacturer. All gel electrophoresis were performed at 180 V for 45 min and stained with SYBR Gold (Invitrogen, Carlsbad, CA, USA). All experiments were performed in triplicate.

### 4.5. The Preparation of Single-Strand Circular (SSC) Templates

The single-stranded DNA template (200 bp, AGGTCGCCAGTGAAGTCTTTCGGGCTTCCTCTTAATCTTTTTGATGCAATCCGCTTTGCTTCTGACTATAATAGTCAGGGTAAAGACCTGATTTTTGATTTATGGTCATTCTCGTTTTCTGAACTGTTTAAAGCATTTGAGGGGGATTCAATGAATATTTATACCGATTCCGCAGTATTGCACTCTATCGTCGCCAGCCC) and a primer (CTGGCGACCTGGGCTGGCGAC) were synthesized by Genscript Biotechnology (Nanjing, China). 100 nM single-stranded DNA template was annealed with an equimolar amount of primer (100 nM) in 1× T4 DNA ligase buffer (New England Biolabs, Ipswich, MA, USA), followed by a stepwise annealing process (95 °C to 25 °C). The annealed products were circularized by incubation with T4 DNA ligase (400 U per reaction, New England Biolabs, Ipswich, MA, USA) at 16 °C overnight, followed by enzyme inactivation at 65 °C for 10 min. To eliminate residual linear DNA, the ligation mixture was digested with Exonuclease I (10 U per reaction, New England Biolabs, Ipswich, MA, USA) and Exonuclease III (100 U per reaction, New England Biolabs, Ipswich, MA, USA) in Exo I buffer at 37 °C for 1 h, and the enzymes were heat-inactivated at 80 °C for 15 min. Circular single-stranded DNA molecules were then purified using the Zymo Oligo Clean & Concentrator Kit (Zymo Research, Irvine, CA, USA) according to the manufacturer’s protocol, quantified with the Qubit ssDNA Assay Kit (Thermo Fisher Scientific Inc., Cleveland, OH, USA) and analyzed by 6% TBE-Urea denaturing PAGE.

### 4.6. Rolling Circle Amplification (RCA)

The primed-template complex was prepared by annealing 800 nM of a phosphorothioate-modified primer (5′-CTGGCGACCTGGGCTGGC*G*A*C-3′) and 200 nM of SSC templates in an annealing buffer (20 mM HEPES, pH 7.5). The mixture was heated to 95 °C for 3 min followed by gradual cooling to room temperature over 25 min.

To evaluate the processivity of DNA polymerases in the presence of different divalent ions, reactions were assembled with 100 nM template, 400 nM primer (forming a primed template at 100 nM), 100 nM polymerase, and 100 μM natural or noncanonical dNTPs. Unless otherwise noted, assays were carried out in buffer 1 (150 mM KCl, 20 mM HEPES, 4 mM TCEP, pH 7.5) supplemented with 1 mM of a single divalent salt (MgCl_2_, MnCl_2_, CaCl_2_, or SrCl_2_). Reactions proceeded for 30 min and were quenched by addition of 0.5 M EDTA. Each assay was repeated thrice.

The unnatural substrates of four kinds were dA6P-Cy3-dT_4_-FldT-dT-FldT-dT_23_-C3, dT6P-Cy3-dT_2_-dSp_8_-dT_20_-C3, dC6P-Cy3-dT_4_-dSp_8_-dT_23_-C3 and dG6P-Cy3-T_30_-C3. They are synthesized according to the previous research [1] (Appendix A).

The RCA products were analyzed by electrophoresis on 0.6% agarose gel. The GeneRuler High Range DNA Ladder (Thermo Fisher Scientific Inc., Cleveland, OH, USA) and 1 kB Plus DNA Ladder (TIANGEN Biotech., Beijing, China) worked as DNA molecular weight standards. A peristaltic pump (BT100-1L; LongerPump Co., Ltd., Baoding, China) circulated the running buffer (46 mM NaCl, 1 mM EDTA, 4 mM NaOH, pH 10) maintain gel temperature. After 2–3 h, the gel was stained with SYBR Gold for 30 min and was analyzed using AzureSpot System (Version 2.0.062, Azure Biosystems, Dublin, CA, USA). The intensity of the RCA products was quantified by Image J (Version 1.48). All experiments were performed in triplicate.

### 4.7. Template Binding Affinity of DNA Polymerases in Various Metal Ion Conditions

Seq99A with a hairpin DNA structure (5′-TTTTTGCGCTCGAGATCTCCGTAAGGAGATCTCGAGCGCGGGACTACTACTGGGATCATCATAGCCACCTCAGCTGCACGTAAGTGCAGCTGAGGTGGC-3′), was used as the primed template for this binding assay (Appendix A) [3]. Before use, Seq99A synthesized by Genscript (Nanjing, China) was heated in a metal bath at 95 °C for 5 min and then returned to room temperature for 30 min. Subsequently, the primed template (100 nM) was incubated with polymerase at a 10:1 molar ratio (10 nM) in 10 μL binding buffer (20 mM HEPES, 4 mM DTT, pH 7.5). To examine ion dependence, reactions were supplemented with 1 mM of either MgCl_2_, MnCl_2_, CaCl_2_, or SrCl_2_ and maintained at 30 °C for 30 min. Then, 10 μL samples and 2.5 μL high-density TBE loading buffers were mixed and run in 4–12% TBE gel (Thermo Fisher Scientific Inc., Cleveland, OH, USA) for 50 min with 120 V condition in an XCell SureLock Mini-Cell (Thermo Fisher Scientific Inc., Cleveland, OH, USA). After staining with SYBR Gold for 30 min, DNA–protein complexes were detected by band mobility shifts and quantified using the AzureSpot imaging system (Version 2.0.062, Azure Biosystems, Dublin, CA, USA) in EPI Blue mode. All experiments were performed three times.

### 4.8. A Pre-Annealed 63/70-mer DNA

To prepare the primed template, a 63-mer oligonucleotide (50 μM, 10 μL; 5′-TACGCAGCGTACATGCTCGTGACTGGGATAACCGTGCCGTTTGCCGACTTTCGCAGCCGTCCA-3′), containing three consecutive phosphorothioate modifications at the 3′ terminus, was mixed with an equimolar amount of its complementary 70-mer (50 μM, 10 μL; 5′-AAACCCTTGGACGGCTGCGAAAGTCGGCAAACGGCACGGTTATCCCAGTCACGAGCATGTACGCTGCGTA-3′) [38] in 80 μL of annealing buffer (1× PBS, pH 7.5, supplemented with 0.01% BSA and 0.05% Tween-20). The mixture was subjected to a thermal cycling program, consisting of denaturation at 95 °C for 5 min followed by slow cooling to ambient temperature to facilitate primed template formation. The resulting primed template was diluted with annealing buffer to a working concentration of 10 μM for subsequent titration assays.

### 4.9. Isothermal Titration Calorimetry (ITC)

To investigate the effect of metal ions on the binding affinity between SS_01 DNA polymerase and primed templates, as well as the thermodynamic profile of subsequent incorporation of dNTPs or modified nucleotides, isothermal titration calorimetry (ITC) was performed using PEAQ-ITC Automated (Microcal, Northampton, MA, USA). All ITC experiments were repeated three times. Data were analyzed using PEAQ-ITC Analysis Software v1.52, assuming a one-site binding model.

#### 4.9.1. Binding of SS_01 to 63/70-mer DNA Under Different Metal Ion Conditions

To test the binding of SS_01 to 63/70-mer DNA under different metal ion conditions, 185 μL of a solution containing 5 μM SS_01 in 1× PBS (pH 7.5) supplemented with 0.01% BSA, 0.05% Tween-20 and no MgCl_2_, 1 mM MgCl_2_, or 1 mM CaCl_2_ was loaded into the sample cell. Then the injection syringe was filled with 40 μL of annealed 63/70-mer DNA at 10 μM or 100 μM, prepared in the same buffer (1× PBS, 0.01% BSA, 0.05% Tween-20). The reference cell was filled with ultrapure water.

Instrument parameters were as follows: temperature = 25 °C; stirring speed = 750 rpm; reference power = 10 μcal/s; feedback mode = High. A total of 19 injections were performed: the first injection of 0.4 μL followed by 18 injections of 2 μL each. The injection rate was set to 0.5 μL/s, with a 150 s interval between injections. Heat changes upon each injection were recorded and integrated to obtain the binding isotherm.

#### 4.9.2. Two-Step ITC Assay for SS_01-Primed-Template Binding Followed by dNTPs or Unnatural Substrate Incorporation

To further analyze the thermodynamics of dNTPs incorporation or modified substrate (single-stranded oligonucleotides modified dNTPs) incorporation, a two-step ITC experiment was performed:

Step 1: Polymerase-63/70-mer primed template binding.

40 μL of pre-annealed 63/70-mer primed template (10 μM in 1× PBS, 0.01% BSA, 0.05% Tween-20) was loaded into the syringe, and 185 μL of 5 μM SS_01 in the same buffer (with no MgCl_2_, 1 mM MgCl_2_, or 1 mM CaCl_2_) was placed in the sample cell. ITC parameters and injection settings were identical to the binding assay described above.

Step 2: Modified nucleotide titration.

Following completion of the binding assay, the resulting mixture containing SS_01 (4.18 μM) and 63/70-mer primed template (1.64 μM) was retained in the sample cell. The syringe was refilled with 40 μL of oligonucleotide-tagged dNTPs solution or dNTPs solution (100 μM in 1× PBS, 0.01% BSA, 0.05% Tween-20). ITC measurements were then performed using the same parameters as in step 1.

The reference cell was again filled with ultrapure water. Heat signals generated upon unnatural dNTPs or dNTPs addition were integrated and analyzed as described above to assess binding or catalytic activity.

#### 4.9.3. Binding Affinity of SS_01 to Unmodified Oligonucleotides or Unnatural Substrates

The binding affinity of SS_01 to oligonucleotide-tagged unnatural dNTP analogs or unmodified single-stranded oligonucleotides was evaluated using the same ITC protocol described for the 63/70-mer binding assay. In these experiments, 185 μL of 5 μM SS_01 in 1× PBS (pH 7.5) containing 0.01% BSA and 0.05% Tween-20 was loaded into the sample cell. The reference cell was filled with ultrapure water.

For unnatural substrates, injection syringes were filled with 40 μL of oligonucleotide-tagged dNTP analogs at three different concentrations: 10 μM, 20 μM, and 100 μM (each prepared in 1× PBS, 0.01% BSA, 0.05% Tween-20). For unmodified single-stranded oligonucleotides, 10 μM oligonucleotide solution in the same buffer conditions were used.

All instrument parameters, injection volumes (0.4 μL for the first injection, 2 μL for the remaining 18 injections or 3 μL for the remaining 12 injections), injection rate (0.5 μL/s), and spacing interval (150 s) remained consistent with previous experiments. Thermodynamic parameters (ΔH, ΔS, and KD) were obtained using nonlinear least-squares fitting of the integrated heat data.

## Figures and Tables

**Figure 1 ijms-26-11909-f001:**
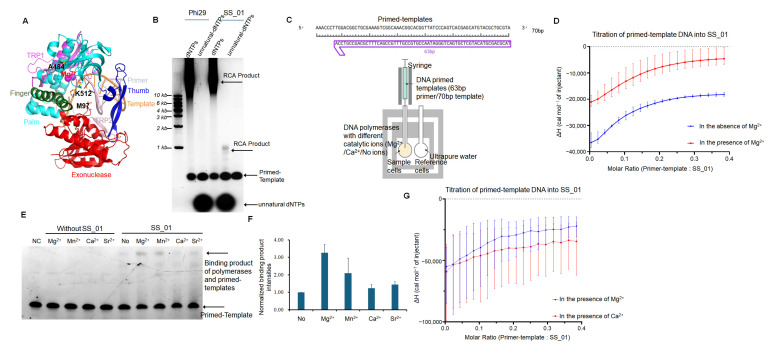
Characterization of SS_01 polymerase activity and primed template binding under different conditions. (**A**): Functional domains and mutation sites of Phi29 DNA polymerase. The exonuclease domain is shown in red, the finger domain in green, the palm domain in cyan, the thumb domain in blue, the TPR2 region in pink, and the TPR1 region in purple. Mg^2+^ ions were indicated as two spheres, and the mutated residues were highlighted in yellow. (**B**): RCA results revealed the processivity of SS_01 using oligonucleotide-tagged dNTPs as substrates. Reactions contained 100 nM primed template, 100 nM polymerase, and 100 μM dNTPs or oligonucleotide-modified dNTPs, incubated at 30 °C for 30 min in Mg^2+^-containing reaction buffer, and analyzed by 0.6% agarose gel electrophoresis. (**C**): Schematic representation of the ITC-based assay used to evaluate binding of the polymerase and primed templates. The sample cell contained 5 μM SS_01 in 1× PBS buffer (pH 7.5) supplemented with 0.01% BSA and 0.05% Tween-20, under three conditions: no MgCl_2_, 1 mM MgCl_2_, or 1 mM CaCl_2_. The injection syringe contained 10 μM annealed 63/70-mer DNA prepared in the same buffer. (**D**): ITC analysis of SS_01 binding to primed templates in the absence or presence of Mg^2+^. The *x*-axis represents the molar ratio of primed template to SS_01 polymerase, and the *y*-axis (ΔH) denotes the enthalpy change per injection, reflecting the heat released or absorbed upon binding. Data are presented as mean ± SD. (**E**): The binding affinity of SS_01 and primed templates under different catalytic ion conditions. A total of 100 nM primed templates was incubated with 10 nM SS_01 in 10 μL binding buffer (20 mM HEPES, 4 mM DTT) supplemented with 1 mM MgCl_2_, MnCl_2_, CaCl_2_, or SrCl_2_ at 30 °C for 30 min. “NC” indicates negative control (primed template only), and “No” indicates reactions without catalytic ions. (**F**): Quantitative analysis of binding affinity for SS_01 and the primed templates. Since no binding bands were observed in the absence of enzyme, we quantified the binding bands present after enzyme addition. Band intensities were measured using Image J (Version 1.48) software and subsequently normalized by the binding band observed under conditions without catalytic ions. (**G**): ITC analysis of SS_01 binding to primed templates in the presence of Mg^2+^ or Ca^2+^. The *x*-axis represents the molar ratio of template to SS_01 polymerase, and the *y*-axis (ΔH) denotes the enthalpy change per injection. Data are presented as mean ± SD.

**Figure 2 ijms-26-11909-f002:**
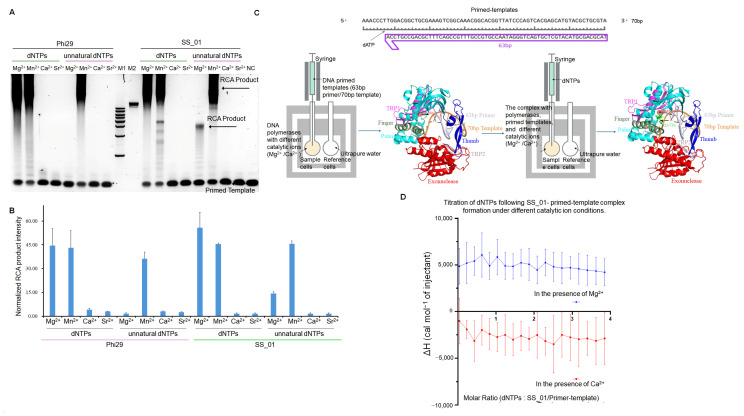
Influence of metal ions on SS_01 polymerase activity and thermodynamics of nucleotide incorporation. (**A**,**B**): Effect of different metal ions on SS_01 polymerase activity with natural dNTPs or oligonucleotide-modified dNTPs. Reactions contained 100 nM primed template, 100 nM polymerase, and 100 μM nucleotides, incubated at 30 °C for 30 min in reaction buffers supplemented with 1 mM of the indicated metal ion. The products were analyzed by 0.6% agarose gel electrophoresis (**A**) and quantified using Image J (**B**). Levels were normalized to the negative control (NC: RCA reagent without polymerases). Data are presented as mean ± SD. (**C**): Schematic representation of the ITC-based assay used to evaluate the incorporation of the polymerase using dNTPs. The ITC experiment was conducted in two steps. In Step 1, 185 μL of SS_01 solution (5 μM in 1× PBS buffer, pH 7.5, supplemented with 0.01% BSA and 0.05% Tween-20) containing 1 mM Mg^2+^ or 1 mM Ca^2+^ was loaded into the sample cell, and 40 μL of annealed 63/70-mer primed templates (10 μM in the same buffer) was titrated from the syringe. In Step 2, the resulting SS_01-primed template complex was retained in the sample cell, and 40 μL of dNTP solution (100 μM in 1× PBS, 0.01% BSA, 0.05% Tween-20) was titrated from the syringe. The reference cell contained ultrapure water. (**D**): ITC analysis of dNTP incorporation by SS_01 using primed templates under different metal ion conditions. The *x*-axis represents the molar ratio of dNTPs to the SS_01-primed template complex, and the *y*-axis (ΔH) denotes the enthalpy change per injection, reflecting the heat released or absorbed upon nucleotide incorporation in the presence of Mg^2+^ or Ca^2+^. Data is presented as mean ± SD.

**Figure 3 ijms-26-11909-f003:**
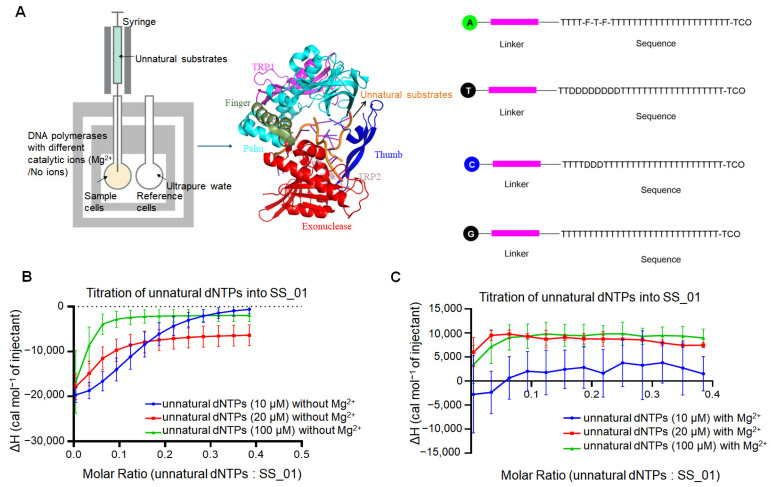
ITC analysis of SS_01 binding to oligonucleotide-modified dNTPs in the absence or the presence of Mg^2+^. (**A**): Schematic representation of the ITC-based assay used to evaluate the binding affinity of polymerases and oligonucleotide-modified dNTPs. The sample cell contained 185 μL of 5 μM SS_01 in 1× PBS buffer (pH 7.5) supplemented with 0.01% BSA and 0.05% Tween-20 in the absence or presence of Mg^2+^, and the reference cell contained ultrapure water. The syringe was filled with 40 μL of oligonucleotide-modified dNTPs at 10, 20, or 100 μM in the same buffer. A, T, C, and G referred to four kinds of oligonucleotide-modified dNTPs instead of natural dNTPs. The structural details of oligonucleotide-modified dNTPs were described in the Materials and Methods section. (**B**): ITC profiles for the binding of SS_01 to oligonucleotide-modified dNTPs in the absence of catalytic metal ions. The *x*-axis shows the molar ratio of oligonucleotide-modified dNTPs to SS_01 polymerase, and the *y*-axis (ΔH) represents the enthalpy change per injection, corresponding to the heat released or absorbed upon binding. Data are presented as mean ± SD. (**C**): ITC profiles for the binding of SS_01 to oligonucleotide-modified dNTPs in the presence of Mg^2+^. The *x*-axis shows the molar ratio of oligonucleotide-modified dNTPs to SS_01 polymerase, and the *y*-axis (ΔH) represents the enthalpy change per injection, corresponding to the heat released or absorbed upon binding. Data are presented as mean ± SD. To enable direct comparison of the heat changes between titrations, the x-axis in our figures uniformly displays the molar ratio calculated under the 10 μM unnatural dNTP condition. For experiments performed at 20 μM and 100 μM, the molar ratios were adjusted by multiplying by the corresponding concentration factors. For example, a molar ratio of 0.2 at 10 μM becomes 0.4 at 20 μM.

**Figure 4 ijms-26-11909-f004:**
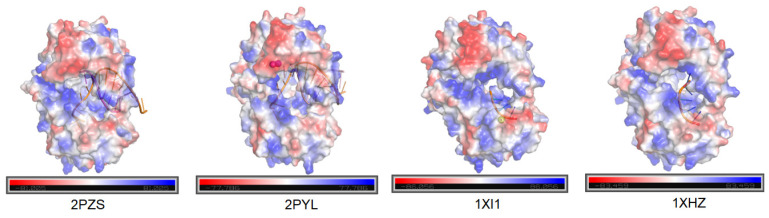
Charged surface analysis of Phi29 DNA polymerases. Surface electrostatic potentials of Phi29 DNA polymerases were analyzed using PyMOL, (Version 2.5.0) with all structures obtained from the PDB database. The analysis indicates that in the presence of Mg^2+^, single-stranded oligonucleotides preferentially enter the exonuclease domain (1XI1 and 1XHZ), whereas double-stranded templates are directed toward the polymerase template-binding site (2PZS and 2PYL).

**Figure 5 ijms-26-11909-f005:**
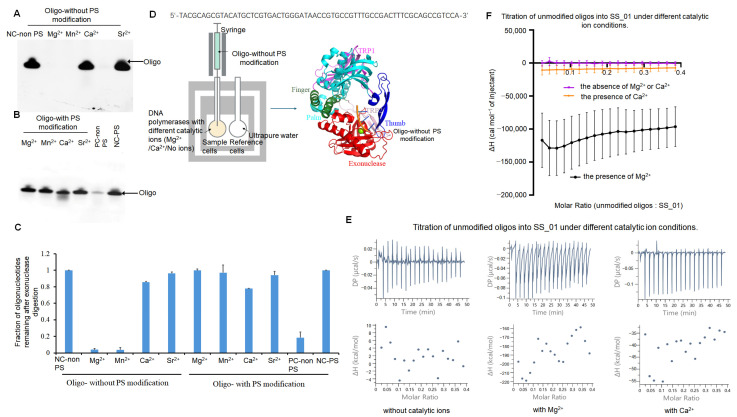
Exonuclease activity and ITC analysis of SS_01 interactions with unmodified oligonucleotides under metal ion conditions. (**A**–**C**): Exonuclease activity assay of SS_01 with unmodified oligonucleotides (non-PS) or phosphorothioate (PS) modified oligonucleotides. Negative control (NC-non PS and NC-PS) referred to the respective oligonucleotides without SS_01. Positive control (PC-non PS) referred to the exonuclease reaction between SS_01 and unmodified oligonucleotides in the presence of Mg^2+^. Reactions were performed with 1 µM oligonucleotides and 200 nM polymerase in 1× phi29 buffer at 30 °C for 30 min, followed by analysis on a 10% TBE–Urea gel (**A**,**B**) and by quantification using Image J (**C**). Levels were normalized to the negative control (NC-PS/NC-non PS). Data is presented as mean ± SD. PS modification effectively protected oligonucleotides from degradation by the exonuclease activity of SS_01. (**D**): Schematic representation of the ITC-based assay used to evaluate the exonuclease activity of polymerases. The sample cell contained 185 μL of 5 μM SS_01 in 1× PBS buffer (pH 7.5) supplemented with 0.01% BSA and 0.05% Tween-20 in the absence or presence of Mg^2+^ or Ca^2+^, and the reference cell contained ultrapure water. The syringe was filled with 40 μL of unmodified oligonucleotides at 10 μM in the same buffer. (**E**): Representative raw ITC results for the interaction between SS_01 and unmodified oligonucleotides under conditions with or without catalytic metal ions. (**F**): ITC data showing the binding of SS_01 to unmodified oligonucleotides in various metal ion conditions. The *x*-axis represents the molar ratio of oligonucleotides to SS_01, and the *y*-axis (ΔH) denotes the enthalpy change per injection. Data are presented as mean ± SD.

**Figure 6 ijms-26-11909-f006:**
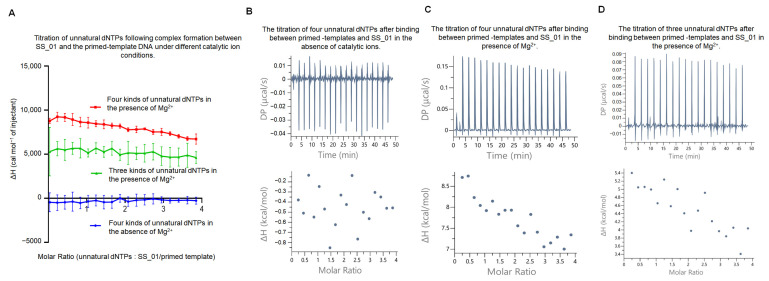
ITC thermodynamic analysis of SS_01 incorporation using oligonucleotide-modified dNTPs. (**A**): ITC data for the incorporation of SS_01 with primed templates and oligonucleotide-modified dNTPs. The ITC experiment was conducted in two steps. In Step 1, 185 μL of SS_01 solution (5 μM in 1× PBS buffer, pH 7.5, supplemented with 0.01% BSA and 0.05% Tween-20) in the absence of presence of Mg^2+^ was loaded into the sample cell, and 40 μL of annealed 63/70-mer primed templates (10 μM in the same buffer) was titrated from the syringe. In Step 2, the resulting SS_01-primed template complex was retained in the sample cell, and 40 μL of unnatural substrates solution (100 μM in 1× PBS, 0.01% BSA, 0.05% Tween-20) was titrated from the syringe. The reference cell contained ultrapure water. The *x*-axis represents the molar ratio of oligonucleotide-modified dNTPs to the SS_01–template complex, and the *y*-axis (ΔH) denotes the enthalpy change per injection, reflecting the heat released or absorbed upon incorporation. (**B**,**C**): Representative raw ITC thermograms showing incorporation of SS_01 with primed templates and oligonucleotide-modified dNTPs under conditions without catalytic metal ions (**B**) or with catalytic metal ions (**C**). (**D**): Representative ITC results for the incorporation of SS_01 with primed templates and three kinds of oligonucleotide-modified dNTPs in the presence of Mg^2+^.

## Data Availability

The data that support the findings of this study are available from the corresponding author upon reasonable request.

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
