# Peer review of "Thermodynamic Profiling Reveals DNA Polymerase Template Binding, Substrate Incorporation, and Exonuclease Function"

_ijms, 2025, doi:10.3390/ijms262411909_

Round 1

Reviewer 1 Report

Comments and Suggestions for Authors

Reviewer:

Recommendation: This paper may be publishable after major revision; I would like to be invited to further review.

Comments:

This manuscript presents a thorough and well-executed investigation of the thermodynamic basis of DNA polymerase template binding, substrate incorporation, and exonuclease function, combining detailed ITC measurements with biochemical assays to provide valuable mechanistic insights; however, certain areas—such as clarity of key findings, quantitative data presentation, and expanded discussion of broader biological implications—would benefit from further refinement to enhance the manuscript’s impact and accessibility for a broad readership.

Overall, the work is novel and addresses a research gap, and with some improvements it will be a valuable contribution to the field.

Regarding specific questions, I would like the authors to address the following points prior to acceptance:

Major comments:

  1. Could the authors further justify their choice of experimental controls in the ITC and exo nuclease activity assays and explain how these controls support their conclusions regarding binding specificity and catalytic requirements?
  2. Have the authors clearly indicated the number of experimental replicates and reported measures of variability (such as standard deviations or errors) for ITC and RCA assays to ensure reproducibility?
  3. Could the authors quantify the bands in the gel-based assays (Figures 1A, 2A, 5A/B) and provide supporting statistical data for observed activity differences?
  4. Do the figure legends, especially for Figures 1, 2, and 3, provide enough detail on experimental conditions—such as buffer compositions and protein/DNA concentrations—for independent interpretation of the results?
  5. Could the authors include a schematic or detailed description of the chemical structure of the oligonucleotide-tagged dNTPs used in this study?
  6. Can the authors elaborate on the mechanistic differences underlying exonuclease activation by different metal ions, particularly between Mg²⁺ and Ca²⁺?
  7. Could the authors provide more specific literature citations, especially when discussing contrasts or similarities to existing knowledge about polymerase and exo nuclease function?

Minor Comments

  1. Ensure consistent use of terms such as “primed-template DNA,” “primed template,” and “duplex DNA” throughout the manuscript to avoid reader confusion.
  2. There are several minor typographical and formatting errors (e.g., missing spaces in “Mg2+” and “Ca2+,” inconsistent italicization of gene/protein names).
  3. Define all abbreviations at first use, e.g., provide full definitions for ITC, RCA, and SSC in the Abstract and Introduction.
  4. Some gel images, especially in Figures 1 and 5, appear low in contrast and may benefit from improved imaging or labeling for clarity.
  5. The statement is present, but it would strengthen the manuscript to specify where data or code can be accessed (e.g., repository link) if possible.

Author Response

Point-by-point response to the Reviewer’s comments:

We thank the reviewers for their time in reviewing our work and for their expert insights and suggestions, which have helped to strengthen the manuscript. Below, please find our detailed point-by-point responses to the specific comments from the reviewers, and we have also revised our manuscript accordingly (green texts). To avoid confusion, we used “New revised Figure. #” to refer to Figures in the revised manuscript and the green shaded sections referred sentences in the revised manuscript.

Reviewer 1:

This manuscript presents a thorough and well-executed investigation of the thermodynamic basis of DNA polymerase template binding, substrate incorporation, and exonuclease function, combining detailed ITC measurements with biochemical assays to provide valuable mechanistic insights; however, certain areas—such as clarity of key findings, quantitative data presentation, and expanded discussion of broader biological implications—would benefit from further refinement to enhance the manuscript’s impact and accessibility for a broad readership.

Overall, the work is novel and addresses a research gap, and with some improvements it will be a valuable contribution to the field.

Regarding specific questions, I would like the authors to address the following points prior to acceptance:

 Major comments:

  1. Could the authors further justify their choice of experimental controls in the ITC and exo nuclease activity assays and explain how these controls support their conclusions regarding binding specificity and catalytic requirements?

Response: We thank the reviewer for the insightful comment regarding the use of experimental controls in the ITC and exonuclease activity assays. To rigorously validate binding specificity and the interpretation of thermodynamic signals, we incorporated several layers of negative controls. We have now also clarified these controls within the Results section of the revised manuscript.

For the ITC experiments assessing polymerase–template binding, we conducted (i) the reaction buffer titrated into the enzyme and (ii) primer–template titrated into the reaction buffer (new revised Supplementary Figure 5 and 6). Both controls produced no detectable ITC data point, in clear contrast to the characteristic binding isotherms observed for the enzyme–template interaction. These results confirm that the measured heats arise from specific complex formation rather than dilution.

For ITC experiments evaluating enzyme interactions with oligonucleotide-tagged dNTPs, we additionally performed oligonucleotide-tagged dNTPs titrated into the reaction buffer, which similarly yielded negligible heat changes (new revised Supplementary Figure 10).

For the exonuclease ITC assay, we performed oligonucleotides titrated into the reaction buffer as a further control. The lack of data point in this condition reinforces that only enzyme–oligonucleotides engagement contributes to the observed thermodynamic signatures (new revised Supplementary Figure 15).

For the incorporation assays involving dNTPs or oligonucleotide-tagged dNTPs by SS_01, we added the reaction buffer titration into the enzyme–primed template complex (new revised Supplementary Figure 9). Unlike the dNTP or oligonucleotide-tagged dNTPs titrations, the reaction buffer titration generated no ITC data point, confirming the specificity of the incorporation-associated heat changes. Additionally, in the ITC assay using oligonucleotide-tagged dNTPs, we performed a control reaction in which the first nucleotide required for extension was omitted. The markedly reduced endothermic heat in this control further demonstrates that the incorporation reaction is intrinsically endothermic and responsible for the observed thermodynamic signal.

Furthermore, in the gel-based exonuclease assays, we included an additional control containing only ions and templates (new revised Figure 1E). The results showed that stronger binding bands appeared only when the enzyme, templates, and Mg²⁺ or Mn²⁺ ions are all present, confirming that the observed bands indeed correspond to specific enzyme–primed templates interactions.

Collectively, these controls substantiate the specificity of binding events and validate the conclusions drawn regarding catalytic requirements in the ITC and exonuclease assays.

  1. Have the authors clearly indicated the number of experimental replicates and reported measures of variability (such as standard deviations or errors) for ITC and RCA assays to ensure reproducibility?

Response: We thank the reviewer for highlighting this important point. We have clearly indicated the number of experimental replicates for all ITC,RCA,polymerase-primedtemplate binding, and exonuclease assays and reported the corresponding measures of variability, ensuring the reproducibility of our results.

The primary ITC experiments were also repeated three times, and the raw titration traces from all three replicates are provided in the Supplementary Figures. ITC control experiments were performed in duplicate with consistent results. All supporting data are included in the Supplementary Information. The reported parameters (ΔH) represent the mean ± standard deviation (SD) of these replicates in new revised Figure 1D, 1G, 2D, 3B, 3C, 5F, 6A.

All gel-based assays (RCA, exonuclease, and binding experiments) were performed in triplicate, as stated in the Methods. In response to the reproducibility, we added statistics analysis using Image J (new revised Figure 1F, 2B, and 5C). These measures of variability provide a quantitative assessment of reproducibility and experimental reliability.

  1. Could the authors quantify the bands in the gel-based assays (Figures 1A, 2A, 5A/B) and provide supporting statistical data for observed activity differences?

ResponseWe thank the reviewer for this valuable suggestion.

Quantitative analyses have been performed. Band intensities were measured using Image J, and the results represent the mean ± standard deviation from three independent experiments. These analyses are now included in the revised manuscript and clearly support the observed differences in polymerase activity.

Regarding original Figure 1A (new revised Figure 1B), we have not provided separate quantification because the data in this panel primarily serve as a qualitative illustration of polymerase activity and substantially overlap with the quantitative results shown in new revised Figure 2A. To avoid redundancy and maintain clarity, we therefore did not duplicate the analysis for original Figure 1A (new revised Figure 1B).

  1. Do the figure legends, especially for Figures 1, 2, and 3, provide enough detail on experimental conditions—such as buffer compositions and protein/DNA concentrations—for independent interpretation of the results?

Response: We appreciate the reviewer’s suggestion. In the revised manuscript, we have updated the figure legends for new revised Figures 1, 2, and 3 to provide clearer descriptions of the experimental conditions—including buffer compositions, ion conditions, and protein/DNA concentrations—so that the results can be more easily and independently interpreted.

  1. Could the authors include a schematic or detailed description of the chemical structure of the oligonucleotide-tagged dNTPs used in this study?

Response: We thank the reviewer for this suggestion. A simplified schematic of the oligonucleotide-tagged dNTPs has already been provided in new revised Figure 3. Detailed chemical structures for each tagged nucleotide are included in new revised Supplementary Figure 17.

  1. Can the authors elaborate on the mechanistic differences underlying exonuclease activation by different metal ions, particularly between Mg²⁺ and Ca²⁺?

Response: We thank the reviewer for this insightful comment. We have expanded the Discussion section to address the mechanistic differences underlying exonuclease activation by different metal ions, particularly between Mg²⁺ and Ca2⁺.

“ITC assays confirmed that degradation was an exothermic process, with stronger heat release in the presence of Mg²⁺ than Ca²⁺, highlighting the essential role of catalytic ions in activating exonuclease function as well as polymerization. This is consistent with previous studies on bacteriophage T4 DNA polymerase, which demonstrated that its exonuclease activity requires two divalent metal ions[35]. One metal ion coordinated with a water molecule to generate a metal–hydroxide species that was positioned to attack the phosphodiester bond at the cleavage site. The second metal ion was proposed to stabilize the leaving 3′-hydroxyl group and to help position the O–P–O bond angles during the transition state.

Our findings were also consistent with previous studies showing that the PabPolB DNA polymerase possessed no exonuclease activity in the presence of Ca²⁺ but retains activity in the presence of Mg²⁺[34]. We speculate that this difference may reflect a mechanism similar to that observed during nucleotide incorporation, in which Ca²⁺ and Mg²⁺ play distinct structural roles. The larger ionic radius of Ca²⁺ compared with Mg²⁺ (1.00 Å vs. 0.72 Å) [18] may result in altered spatial positioning within the active sites. Further structural studies under different ionic conditions will be necessary to fully elucidate the mechanistic basis of SS_01 exonuclease activity.”

  1. Could the authors provide more specific literature citations, especially when discussing contrasts or similarities to existing knowledge about polymerase and exo nuclease function?

Response: We thank the reviewer for this valuable suggestion. In the revised manuscript, we have expanded the Discussion section to provide more detailed comparisons with previously reported findings on DNA polymerases and exonuclease mechanisms. We have also incorporated additional literature citations to support these discussions. These revisions offer a clearer and more comprehensive interpretation of our experimental observations.

Minor Comments

  1. Ensure consistent use of terms such as “primed-template DNA,” “primed template,” and “duplex DNA” throughout the manuscript to avoid reader confusion.

Response: We thank the reviewer for this suggestion. We have ensured consistent terminology throughout the manuscript, and all instances have been changed to “primed template” for clarity.

  1. There are several minor typographical and formatting errors (e.g., missing spaces in “Mg2+” and “Ca2+,” inconsistent italicization of gene/protein names).

Response: We thank the reviewer for pointing this out. We have checked the manuscript and corrected the formatting of “Mg²⁺” and “Ca²⁺”. Only species names (vibrio parahaemolyticus phage VpV262 DNA polymerases, psychrobacillus PB DNA polymerases, and E.coli) and subheadings have been italicized for consistency.

  1. Define all abbreviations at first use, e.g., provide full definitions for ITC, RCA, and SSC in the Abstract and Introduction.

Response: We thank the reviewer for this suggestion. In the revised manuscript, ITC is written full in the Abstract. ITC and RCA are defined at their first appearance in the Introduction. SSC is defined at its first occurrence in the Result section. All abbreviations are now clearly introduced where they first appear.

  1. Some gel images, especially in Figures 1 and 5, appear low in contrast and may benefit from improved imaging or labeling for clarity.

Response: We thank the reviewer for this comment. We have replaced the gel images with higher-resolution versions obtained directly from the gel documentation system. For the binding assay in new revised Figure 1E, due to the intrinsic properties of the gel, we have selected the clearest representative image available to best illustrate the results.

  1. The statement is present, but it would strengthen the manuscript to specify where data or code can be accessed (e.g., repository link) if possible.

Response: We thank the reviewer for the suggestion. All ITC raw data from replicate experiments have been provided in the Supplementary Figures.

Reviewer 2 Report

Comments and Suggestions for Authors

The paper by Yaping Sun et al. is dedicated to studying the Phi29 polymerase mutant with various substrates in the presence of different cations, as well as the influence of modified nucleic acids.

The work is interesting, but the presentation of the material does not give the reader an understanding of what was actually done by the authors of this work. Furthermore, it is necessary to confirm the conclusions made in the work.

The work requires major revision.

The most significant comments on the work are the following.

1) When conducting experiments using the ITC method, control experiments are necessary to confirm that the observed heat of reaction corresponds to the process of interaction of substance A from the syringe with substance B in the cell. For this, in each case, it is necessary to perform a titration of substance A into a cell with buffer, and an experiment of adding buffer from the syringe into a cell with substance B is also required.

2) Data obtained by the ITC method should be presented in the main text and in the SI. All titration curves (heat flow over time), as well as individually processed curves (values of the heat effect versus the protein/ligand ratio) at three repeat must be provided.

3) The data presented in figure 2 D is typical for an ITC experiment with different buffer concentrations in the syringe and cell. Moreover, in many other titration graphs, a linear dependence on the amount of added substance is observed, which is typical for the case when the experiment has different buffer concentrations in the cell and syringe, associated with the heat of hydration of ions and substances in the buffer. The authors must clearly separate the effects associated with protein interaction from the artifact described above.

4)  The data presented in Fig. 1C on the binding of the enzyme to DNA require a control experiment without protein, but with the addition of various types of ions. Only in this case can one state that the marked band on the electropherogram corresponds to the product of polymerase binding to DNA.

5) When reading the work, it is very difficult to understand which model systems were used to conduct the experiments. I strongly recommend that in each section with different compounds interacting with the polymerase, describe what these models are, their length, composition (with reference to the methods section). Furthermore, it would be very useful to provide a figure of the model systems, as well as various cofactors (ions, dNTP, modified dNTP). I recommend introducing a designation (few symbols) for each model for easier perception of the publication material.

6) It remains unclear to me what modified dNTP is. Are these the substrates described on page 13, line 479? For these substrates, it is necessary to provide the chemical structure, and not just refer to another publication.

7) The authors should provide the values of the thermodynamic parameters ΔH⁰, ΔS⁰, ΔG⁰(25°C) and KD (the D should be subscript), as well as the mixing stoichiometry for all studied reactions where it was possible to determine these values, in the form of a table. For each value, it is necessary to provide the error magnitude.

8) Line 121-123: it is necessary to process the electrophoresis data quantitatively and draw conclusions based on quantitative comparison.

9)  Line 135-136: “Ca²⁺ and Sr²⁺—with significantly larger radii—act as negative controls, as their geometric mismatch disrupts active-site coordination and prevents catalysis.” This is just a speculation without any experimental confirmation.

10) Line 224-227 “When 100μM primed-template DNA was used in the presence of Mg²⁺ conditions, the binding reaction rapidly reached equilibrium, resulting in negligible overall exothermic changes (Supplementary Figure 1A).” This figure lacks confirmation of rapid saturation of the protein with the product. The "drop" of the first point in the titration curves is common and not a pattern in the observed titration dependence, and in most cases it is not considered during analysis.

11) Why in Fig. 5 was titration carried out only up to a ratio of 0.4? This range does not allow determining the quantitative characteristics of the protein-DNA interaction.

12) In the materials and methods section, the authors should provide a more detailed description of how electrophoresis was performed, indicate what equipment was used for ITC, and what software and physico-chemical model were used to process the experimental data.

The authors should carefully proofread the text. There are undescribed or poorly described aspects, for example:

13) What is the SS_01 mutant? One can assume it is mentioned in line 81.

14) Line 250 – it is unclear what VpV62 is.

15) It is unclear what Tag is, for example, in Line 549.

Author Response

Point-by-point response to the Reviewer’s comments:

We thank the reviewers for their time in reviewing our work and for their expert insights and suggestions, which have helped to strengthen the manuscript. Below, please find our detailed point-by-point responses to the specific comments from the reviewers, and we have also revised our manuscript accordingly (green texts). To avoid confusion, we used “new revised Figure. #” to refer to Figures in the revised manuscript and the green shaded sections referred sentences in the revised manuscript.

Reviewer 2

The paper by Yaping Sun et al. is dedicated to studying the Phi29 polymerase mutant with various substrates in the presence of different cations, as well as the influence of modified nucleic acids.

 The work is interesting, but the presentation of the material does not give the reader an understanding of what was actually done by the authors of this work. Furthermore, it is necessary to confirm the conclusions made in the work.

The work requires major revision.

The most significant comments on the work are the following.

1) When conducting experiments using the ITC method, control experiments are necessary to confirm that the observed heat of reaction corresponds to the process of interaction of substance A from the syringe with substance B in the cell. For this, in each case, it is necessary to perform a titration of substance A into a cell with buffer, and an experiment of adding buffer from the syringe into a cell with substance B is also required.

Response:We thank the reviewer for this important suggestion. In response, we have performed a series of negative control ITC experiments to confirm that the heat changes observed in the main experiments are derived from specific interactions rather than from buffer dilution or mixing effects.

We found that most buffer titrations produced negligible heat signals, resulting in no discernible data points. The corresponding control data are now included in the Supplementary Information.

Binding of SS_01 to primed-template DNA:

We conducted control titrations by (i) injecting the reaction buffer into the enzyme solution and (ii) titrating primed-template DNA into the reaction buffers containing different catalytic metal ions. Each control was performed in duplicate, and in all cases, no valid data points could be obtained. The results are shown in the new revised Supplementary Figure 5 and 6.

Binding of SS_01 to unnatural dNTPs under Mg²⁺ conditions:

As the titration of the reaction buffer into SS_01 were already included in the experiments above, here we performed titrations of unnatural dNTPs into the reaction buffer containing different catalytic ions (new revised Supplementary Figure 10). The results were consistent with those from the enzyme–template binding controls—no data points were generated, and both replicates gave identical outcomes.

Exonuclease activity assays with non-modified oligonucleotides:

We also performed control titrations of non-modified oligonucleotides into the reaction buffer (new revised Supplementary Figure 15). Again, no data points were observed.

Polymerization assays involving dNTPs or unnatural nucleotide substrates:

We carried out control titrations of the reaction buffer into the enzyme–primed-template complex. Following our two-step ITC protocol, we first established the enzyme–template complex and then replaced the substrate solution with the reaction buffer for titration (new revised Supplementary Figure 9). Consistent with the other controls, these titrations also failed to generate detectable data points.

Together, these negative control experiments confirm that the measurable heat changes in our main ITC assays are attributable to specific substrate–enzyme interactions rather than buffer-related effects.

2) Data obtained by the ITC method should be presented in the main text and in the SI. All titration curves (heat flow over time), as well as individually processed curves (values of the heat effect versus the protein/ligand ratio) at three repeat must be provided.

Response:We thank the reviewer for this constructive suggestion. As recommended, we have now included the complete ITC raw data for all experiments in the supplementary figures to ensure full transparency.

For the main ITC experiments, the supplementary figures now contains the titration curves (heat flow over time) and the corresponding binding isotherms (heat effect versus molar ratio) from three independent replicates.

For the control experiments, we have performed two independent replicates. As these two replicates yielded nearly identical results, confirming the reliability of the results, both are included in the supplementary figures.

3) The data presented in figure 2 D is typical for an ITC experiment with different buffer concentrations in the syringe and cell. Moreover, in many other titration graphs, a linear dependence on the amount of added substance is observed, which is typical for the case when the experiment has different buffer concentrations in the cell and syringe, associated with the heat of hydration of ions and substances in the buffer. The authors must clearly separate the effects associated with protein interaction from the artifact described above.

Response: We thank the reviewer for raising this important point regarding the potential for the buffer mismatch artifacts in ITC experiments. We agree that such effects must be carefully ruled out. Please allow us to clarify why the signals we observe are attributable to specific protein-ligand interactions rather than buffer-related artifacts.

First, all our ITC experiments were performed under meticulously matched buffer conditions. The identical buffer (1× PBS supplemented with 0.01% BSA and 0.05% Tween-20) was used in both the syringe and the sample cell, thereby eliminating the possibility of heat effects arising from differences in buffer composition or ion hydration.

Second, and more critically, the distinct thermodynamic signatures provided direct evidence against a nonspecific buffer artifact. As presented for direct comparison in the original Figure 2C and 2D (now new revised Supplementary Figures 7 and 8), the heat flow in the presence of Mg²⁺ is endothermic, while it is exothermic in the presence of Ca²⁺. The fact that these signals are opposite in direction—despite identical buffer conditions—strongly indicates that they originate from distinct, ion-dependent biochemical events at the protein's active site.

Finally, the robustness of our data is further supported by the fact that all key ITC measurements were conducted as three independent biological replicates, using proteins from separate purifications and performed at different times. The consistent results across these replicates underscore the reliability of our findings.

Therefore, our thermal changes reported, particularly in original Figure 2D, reflect genuine ion-dependent enzymatic interactions

4)  The data presented in Fig. 1C on the binding of the enzyme to DNA require a control experiment without protein, but with the addition of various types of ions. Only in this case can one state that the marked band on the electropherogram corresponds to the product of polymerase binding to DNA.

Response: We appreciate the reviewer’s insightful comment. To address this concern, we have performed additional control experiments in which no protein was added, but various types of metal ions (Mg²⁺, Mn²⁺, Ca²⁺, etc.) were included under the same conditions as in new revised Figure 1E. The results showed that in the absence of polymerase, the addition of ions alone did not lead to any shifted bands of templates on the electrophoretic mobility shift assay gel. This confirms that the shifted band observed in new revised Figure 1E indeed corresponds to the specific binding of the polymerase to DNA, rather than an artifact caused by ionic effects.

5) When reading the work, it is very difficult to understand which model systems were used to conduct the experiments. I strongly recommend that in each section with different compounds interacting with the polymerase, describe what these models are, their length, composition (with reference to the methods section). Furthermore, it would be very useful to provide a figure of the model systems, as well as various cofactors (ions, dNTP, modified dNTP). I recommend introducing a designation (few symbols) for each model for easier perception of the publication material.

Response: We thank the reviewer for this helpful suggestion. To improve clarity, schematic representations of the model systems, including their sequences and the various cofactors (ions, dNTPs, and modified dNTPs), have been provided in Figures 1C, 2C, 3A, and 5D. These figures include clear designations for each model to facilitate reader understanding and interpretation of the experimental data.

6) It remains unclear to me what modified dNTP is. Are these the substrates described on page 13, line 479? For these substrates, it is necessary to provide the chemical structure, and not just refer to another publication.

Response: We thank the reviewer for this suggestion. A simplified schematic of the oligonucleotide-tagged dNTPs has already been provided in new revised Figure 3. Detailed chemical structures for each tagged nucleotide are included in new revised Supplementary Figure 17.

7) The authors should provide the values of the thermodynamic parameters ΔH⁰, ΔS⁰, ΔG⁰(25°C) and KD (the D should be subscript), as well as the mixing stoichiometry for all studied reactions where it was possible to determine these values, in the form of a table. For each value, it is necessary to provide the error magnitude.

Response: We thank the reviewer for pointing this out. We have now included a comprehensive summary of the thermodynamic parameters in new revised Supplementary Table 1 and 2. This table reports the KD, ΔH, ΔS, and ΔG (at 25°C) for all quantified interactions, with errors presented as the standard deviation from three independent replicates.

8) Line 121-123: it is necessary to process the electrophoresis data quantitatively and draw conclusions based on quantitative comparison.

ResponseWe thank the reviewer for this valuable suggestion. Quantitative analyses have been performed. Band intensities were measured using Image J, and the results represent the mean ± standard deviation from three independent experiments. These analyses are now included in the revised manuscript and clearly support the observed differences in polymerase activity.

9)  Line 135-136: “Ca²⁺ and Sr²⁺—with significantly larger radii—act as negative controls, as their geometric mismatch disrupts active-site coordination and prevents catalysis.” This is just a speculation without any experimental confirmation.

ResponseWe thank the reviewer for this comment. We would like to clarify that our statement regarding the use of Ca²⁺ and Sr²⁺ as negative controls is supported by experimental evidence in the literature.

The ionic radii data cited are from “Shannon, R.D., Revised effective ionic radii and systematic studies of interatomic distances in halides and chalcogenides, Acta Cryst, 1976, A32: 751–767.”

Furthermore, both Ca²⁺ and Sr²⁺ have been previously employed in nanopore sequencing experiments to generate characteristic blockade currents as sequencing signals (references 19 and 20:

19 Nikiforov, T.T., Oligonucleotides labeled with single fluorophores as sensors for deoxynucleotide triphosphate binding by DNA polymerases. Anal Biochem, 2014. 444: p. 60-6.

  1. Stranges, P.B., et al., Design and characterization of a nanopore-coupled polymerase for single-molecule DNA sequencing by synthesis on an electrode array. Proc Natl Acad Sci U S A, 2016. 113(44): p. E6749-E6756.)

In these conditions, DNA polymerases can bind the ions but are unable to extend the DNA, allowing observation of the first base incorporation without elongation. This is further supported by studies showing that substitution of Mg²⁺ with Ca²⁺ in DNA polymerase β leads to a slower rate of phosphodiester bond formation and reduced nucleotide selectivity, indicating that the larger ionic radius of Ca²⁺ impairs catalytic efficiency (Ralec C, Henry E, Lemor M, Killelea T, Henneke G. Calcium-driven DNA synthesis by a high-fidelity DNA polymerase. Nucleic Acids Res. 2017 Dec 1;45(21):12425-12440. doi: 10.1093/nar/gkx927. PMID: 29040737; PMCID: PMC5716173.).

Structural studies have also shown that Ca²⁺ can occupy the catalytic center of DNA polymerases, providing further mechanistic insight into how these ions act as non-productive cofactors (Chang, C., Lee Luo, C. & Gao, Y., Nat Commun 13, 2346, 2022; https://doi.org/10.1038/s41467-022-30005-3.). Therefore, the choice of Ca²⁺ and Sr²⁺ as negative controls is grounded in previously reported experimental results, rather than mere speculation.

We have also added more explanation in the discussion section to provide further clarity.

10) Line 224-227 “When 100μM primed-template DNA was used in the presence of Mg²⁺ conditions, the binding reaction rapidly reached equilibrium, resulting in negligible overall exothermic changes (Supplementary Figure 1A).” This figure lacks confirmation of rapid saturation of the protein with the product. The "drop" of the first point in the titration curves is common and not a pattern in the observed titration dependence, and in most cases it is not considered during analysis.

ResponseWe thank the reviewer for this comment. We would like to clarify that the conclusion of rapid equilibrium in original Supplementary Figure 1A (new revised Supplementary Figure 13) was not based on the first data point of the titration. When titrating with 10 μM primed template, a clear titration response is observed, with the heat change gradually decreasing as more DNA is added. In contrast, when 100 μM primed template is used, excluding the first point, the subsequent heat changes (ΔH values) are very small, and no clear trend of decreasing heat with additional injections is observed. This indicates that the protein is rapidly saturated under these conditions, and the negligible overall exothermic change reflects true equilibrium rather than an artifact of the initial point.

To avoid any potential ambiguity, we have removed this speculative statement from the manuscript.

“When 100 μM primed template was used in the presence of Mg2+ conditions, the binding reaction revealed negligible overall exothermic changes”

11) Why in Fig. 5 was titration carried out only up to a ratio of 0.4? This range does not allow determining the quantitative characteristics of the protein-DNA interaction.

Response: We thank the reviewer for this comment. We would like to clarify that the titration in original Figure 5D (new revised Figure 5F) was performed up to a protein-to-DNA ratio of 0.4 based on prior experiments (new revised Figure 5A), which demonstrated that 200 nM of enzyme with 1 μM of unmodified oligonucleotide (enzyme : oligos = 1:5, corresponding to a ratio of 0.2) reaches full reaction in the presence of Mg2+. Namely, under Mg²⁺ and Mn²⁺, the unmodified oligonucleotide is completely degraded.

Therefore, for the subsequent ITC experiments, we focused on analyzing the heat changes associated with exonuclease activity. Under Mg²⁺ conditions, a clear exothermic response is observed. Since the reaction is essentially complete by the time the ratio reaches 0.2, extending the titration to higher ratios would not provide additional quantitative information about the binding interaction, but rather reflect the complete enzymatic reaction.

12) In the materials and methods section, the authors should provide a more detailed description of how electrophoresis was performed, indicate what equipment was used for ITC, and what software and physico-chemical model were used to process the experimental data.

Response:Thank you for the reviewer’s constructive comment. We have revised the Materials and Methods section to include detailed descriptions of the electrophoresis conditions used in the exonuclease assay, RCA reaction, and DNA-binding experiments. Furthermore, we have specified the ITC instrument model as well as the software and physico-chemical model applied for data fitting and analysis.

The authors should carefully proofread the text. There are undescribed or poorly described aspects, for example:

13) What is the SS_01 mutant? One can assume it is mentioned in line 81.

Response: We thank the reviewer for pointing this out. We have added a clear explanation of the SS_01 mutant in the main text, specifying its origin and relevant characteristics.

14) Line 250 – it is unclear what VpV62 is.

Response: We thank the reviewer for the comment. We have added a clear explanation of VpV62 in the main text.

15) It is unclear what Tag is, for example, in Line 549

Response: We thank the reviewer for this valid point. Indeed, "Tag" was used in the figures as an abbreviation for "oligonucleotide-modified dNTPs," which was defined in the corresponding legends.

In light of the comment, and to prevent any potential misunderstanding, we have now replaced "Tag" throughout the manuscript and figures with the more explicit terms "unnatural dNTPs".

Round 2

Reviewer 1 Report

Comments and Suggestions for Authors

Reviewer:

Recommendation: This manuscript is publishable as it is. 

Comments:

The authors have made significant corrections at appropriate parts of the manuscript. I really appreciate their efforts. I hope it will be published in its current form.